# Insight into the introduction of domestic cattle and the process of Neolithization to the Spanish region Galicia by genetic evidence

Marie Gurke[1], Amalia Vidal-Gorosquieta[2], Johanna L. A. Pajimans[3], Karolina Węcek[4], Axel Barlow[5], Gloria González-Fortes[6], Stefanie Hartmann[1], Aurora Grandal-d'Anglade[2], Michael Hofreiter[1]*

1 Institute of Biochemistry & Biology, University of Potsdam, Potsdam, Germany, 2 Instituto Universitario de Xeoloxía, Universidade da Coruña, A Coruña, Spain, 3 Department of Genetics & Genome Biology, University of Leicester, Leicester, United Kingdom, 4 Department of Comparative Anatomy, Institute of Zoology and Biomedical Research, Jagiellonian University, Kraków, Poland, 5 School of Science and Technology, Nottingham Trent University, Nottingham, United Kingdom, 6 Department of Life Science and Biotechnology, University of Ferrara, Ferrara, Italy

☯ These authors contributed equally to this work.
* michael.hofreiter@uni-potsdam.de

**Data Availability Statement:** The mitochondrial genomes in our manuscript have the GenBank accession numbers MW689247 to MW689255

## Abstract

Domestic cattle were brought to Spain by early settlers and agricultural societies. Due to missing Neolithic sites in the Spanish region of Galicia, very little is known about this process in this region. We sampled 18 cattle subfossils from different ages and different mountain caves in Galicia, of which 11 were subject to sequencing of the mitochondrial genome and phylogenetic analysis, to provide insight into the introduction of cattle to this region. We detected high similarity between samples from different time periods and were able to compare the time frame of the first domesticated cattle in Galicia to data from the connecting region of Cantabria to show a plausible connection between the Neolithization of these two regions. Our data shows a close relationship of the early domesticated cattle of Galicia and modern cow breeds and gives a general insight into cattle phylogeny. We conclude that settlers migrated to this region of Spain from Europe and introduced common European breeds to Galicia.

## Introduction

Neolithization is the transition of human societies from hunting and gathering to settlement and agriculture [1]. During this process, which started around 10,500 BP in the Middle East, selected plants and animals were domesticated by humans for food, clothing, or other resources these species provided [2]. Genetic data of ancient and modern samples from domestic cattle *Bos taurus* suggest that the domestication of its wild ancestor, the aurochs *Bos primigenius*, also began in the Middle East [3]. Domestic cattle spread from the Middle East to Anatolia, Europe and Africa, and local domestication or maternal interbreeding with *B.*

(full details below). The original Illumina reads are available in the short read archive (SRA) under accession number PRJNA705960. Full details of GenBank submission, including accession numbers: BankIt2434488 Bt_AR_1200BP MW689247 BankIt2434488 Bt_AR_2460BP MW689248 BankIt2434488 Bp_CL_9100BP_1 MW689249 BankIt2434488 Bp_CL_9100BP_2 MW689250 BankIt2434488 Bp_CL_9100BP_3 MW689251 BankIt2434488 Bt_AT_3370BP MW689252 BankIt2434488 Bt_AT_1600BP MW689253 BankIt2434488 Bt_SA_1290BP_2 MW689254 BankIt2434488 Bt_SA_1200BP_1 MW689255.

**Funding:** Grant CGL2014–57209-P awarded to AGD Spanish Ministry of Science and Innovation (https://www.ciencia.gob.es/portal/site/MICINN/menuitem.7eeac5cd345b4f34f09dfd1001432ea0/?vgnextoid=f1e81f4368aef110VgnVCM1000001034e20aRCRD) and Grant CGL2014–57209-P awarded to AGD Xunta de Galicia (https://www.edu.xunta.gal/portal/es) The funders had no role in study design, data collection and analysis, decision to publish, or preparation of the manuscript.

**Competing interests:** The authors have declared that no competing interests exist.

*primigenius* seems to have had little effect on today's gene pool [3, 4], although there is clear evidence that it occasionally happened [5–12].

Nowadays, all domestic cattle belong to four major mitochondrial haplogroups, T, Q, R, and I. Haplogroup Q originated together with T in the Middle East, while haplogroup R has so far only been found in Italy; it has been argued to be either the product of maternal interbreeding of European aurochs or to go back to a second independent domestication event [7, 13, 14]. However, a recent paleogenome study found an R haplotype in an almost 9,000 year old *B. primigenius* sample from Morrocco [14], leaving its origin open to discussion. A separate domestication event in southern Asia gave rise to zebu or indicine cattle, *B. indicus*, and this group is assigned to haplogroup I [15, 16]. Based on mitochondrial DNA data, a founding population of 29 to 783 females has been estimated for the beginning of the taurine cattle domestication in the Middle East [4, 17]. Following this bottleneck at the onset of the Neolithic, domestic stocks spread across East Africa, Anatolia, and Europe in co-migration with humans. Most of these taurine cattle nowadays belong to six sub-haplogroups of the mitochondrial haplogroup T [3]. Sub-haplogroups T1, T2, and T3 are derived from ancestral haplogroup T, and today sub-haplogroup T3 is dominant across the European mainland and Britain, while T1 mainly occurs in Africa [3]. T2 is currently common in Europe and the Middle East, but less frequent than T3 in Europe. The remaining sub-haplogroups have a much more narrow distribution, with T4 restricted to Asian breeds [6], T5 to Italian breeds [7], and T6 has only survived in a few Balkan breeds [9].

A region where the European mainland and Africa are in close proximity to each other is the Iberian Peninsula, to which pioneering farmers may have arrived from both the African and European coasts of the Mediterranean sea [18, 19]. Indeed, ancient domestic cattle are thought to have reached Spain at two separate times and following two separate routes. From the Near East, taurine cattle spread north-westward into Anatolia and south-westward into North Africa [14, 20]. From Anatolia, the early domestic cattle expanded into Europe following two different ways: One, associated with the LBK (Linearbandkeramik) culture and running along the Danube river, the other along the Mediterranean coast, associated with the Cardial culture and likely involving a smaller number of individuals transported by boat. In agreement with this hypothesis, there is a cline of genetic diversity from Eastern to Central-Western Europe, with the Iberian and Southern French cattle being the ones showing the lowest genetic diversity within Europe. Later on, African taurine cattle were probably introduced into Iberia from the Maghreb, from where they expanded into Southern Europe with a subsequent increase of the African haplogroup T1 in these areas [21]. Although the Iberian Peninsula has been at the center of studies trying to elucidate the times and ways of expansion of the Neolithic into Europe [22–24], most of the efforts have focused on the Northeastern and Mediterranean regions, while less is known for the Northwest or Atlantic Spain [18]. In the Cantabrian region, which includes the northern coast, Neolithization is dated to 8,000-4,000 BP, and there it is the result of a complex mosaic process in which hunters and gatherers lived along with settlers [25]. The first domestic animals of this region date to around 7,000 BP and were mostly ovicaprines, which were later followed by pigs and cattle [1]. However, much less is known about the region adjacent to Cantabria, Galicia, because no Neolithic sites have been identified in this region yet [26]. The oldest genetic information for Galicia is from the Chalcolithic period. It dates to 5,950-5,050 years BP and therefore to a time period in which settlers with domestic livestock were much more common than other forms of human societies [26]. To address this gap of knowledge and gain insights into the process of Neolithization in Galicia, we analyzed the mitochondrial DNA from 18 samples of cattle mostly found in mountain caves in this region, of which eleven yielded enough DNA for phylogenetic analysis. Their ages range from before the Neolithic up to recent (last 60 years), and they include the domestic

cattle *B. taurus* as well as its pre-Neolithic wild ancestor *B. primigenius*. We used these data to retrace the genetic development of cattle in this area and compare them to their European, African, middle Eastern, and Asian relatives.

## Materials and methods

Bone samples were collected from nine sites in Galicia, which are shown in Fig 1. Two of them were archaeological sites, but most of the bones were found in mountain caves, that were natural traps for the cattle. We tentatively assigned them to species based on their morphology and determined the age of the bones by direct $^{14}$C dating or by $^{14}$C dating of other bones found in the same cave (stratigraphic). All specimens are housed in the "Instituto Universitario de Xeoloxía, Universidade da Coruña", the director of which is one of the co-authors (Aurora Maria Grandal D'Anglade). Therefore, no permits were required. For all samples, DNA was extracted from bone powder following a published protocol [27]. Genetic libraries were prepared from

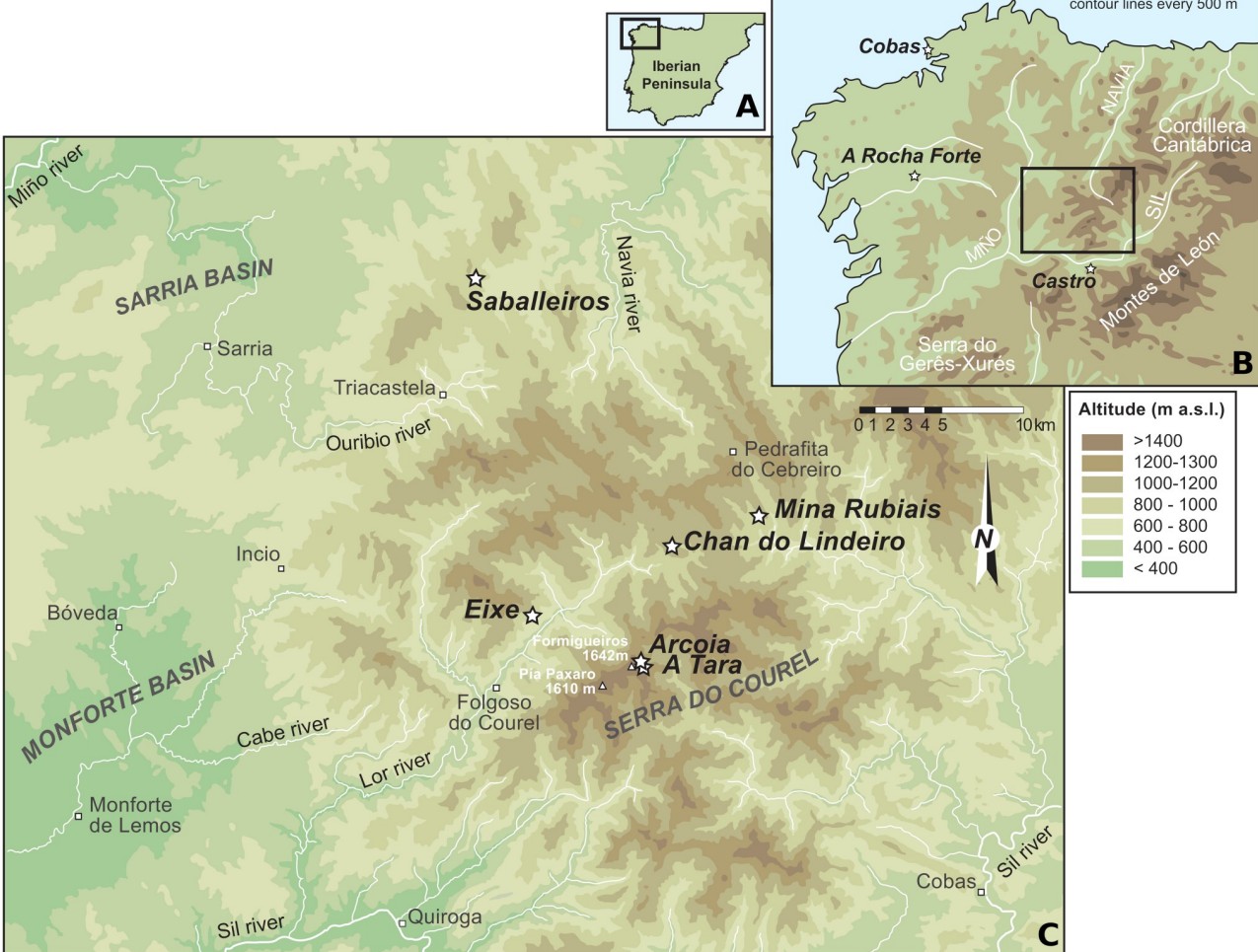

**Fig 1. Map of Galician mountain caves.** A. The black rectangle shows the location of Galicia within the Iberian Peninsula. B. Map of Galicia, corresponding to the black rectangle region in A. C. More detailed map of the region corresponding to the black rectangle in B); it shows the location of caves in which bone samples described in this study were found; stars denote cave locations. (The map was originally created by author Aurora Grandal-d'Anglade specifically for this study.).

10 $\mu$l (for the samples from Cobas, A Rocha Forte, Minas Rubiais, and A Tara) or 20 $\mu$l (for the remaining samples) extract following a single-stranded library preparation protocol [28]. Uracil-DNA glycosylase (UDG) was used to remove uracil. To enrich the resulting libraries for mitochondrial DNA, hybridization capture was performed according to a solid-state capture protocol, using the Agilent SureSelect 244k MicroArray. The baits on the array were designed based on the complete mitochondrial genome from an aurochs (*B. primigenius*; GenBank Acc Nr NC_013996.1) using 60 bp bait length and 3 bp tiling space. Hybridization capture was carried out in two consecutive capture rounds according to a published protocol [29]. The resulting product was then sequenced on the Illumina NextSeq 500 platform using 75 bp paired-end chemistry.

Sequence reads were adapter-trimmed and quality-filtered at a phred quality score cutoff of 28 using cutadapt-1.12 [30], forward and reverse reads were merged with flash-v-1.2.10 [31] were possible, and reads below a length of 30 bp were excluded. Mapping of merged reads against a *B. taurus* mitochondrial reference genome (GenBank Acc Nr.: AY526085) was done with the program bwa-v0.7.8 aln with default parameters [32]. We filtered out all alignments with a quality score below 30 using samtools-v0.1.19 (program: view) [33]. The same samtools version was used to remove duplicates (program: rmdup) and to collect statistics about the alignments (programs: index, idxstats, and depth). We used mapDamage 2.0.7 to assess damage patterns of the ancient DNA samples [34] and computed consensus sequences based on a minimum depth of 3 reads with angsd-v0.914 using the majority rule [35]. Consensus sequences were checked manually. If possible, samples were assigned to haplogroups based on diagnostic single nucleotide polymorphisms (SNPs) in the D-Loop region of the mitochondrial genome of *B. taurus* as they are described in previous studies [3, 6, 9, 36].

With the consensus sequences, we generated three multiple sequence alignments using MAFFT v7.310 and its FFT-NS-i algorithm with 1000 iterations [37]. All alignments were checked by eye for any obvious misalignments. First, we generated a multiple sequence alignment that contained data from most common cattle haplogroups and included all of our samples. As references, we used mitochondrial genome sequences of the species *Bos taurus*, *Bos indicus*, *Bos grunniens* and *Bos primigenius* from a published study [38]. The complete dataset then covered five cattle haplogroups (T, R, I, P, Q) and five sub-haplogroups of T (T1, T2, T3, T4, T5). The resulting alignment was used for a Bayesian phylogenetic analysis with the program BEAST v1.10.4 [39]. For this, we transferred the annotation of the reference mitochondrial genome EU177854 to our alignment, and starts and ends of the features were moved according to gaps in the alignment. Then, we predicted the best models of molecular evolution for each of the features in the annotation with the program partitionfinder-2.1.1 [40]. The analysis with BEAST was done using the best model predicted for each partition, a strict molecular clock, and a coalescent model, assuming a constant population size. It ran with a chain length of 100 million and a sampling every 1,000 generations. 10% of the generations were used as a burn-in, and the convergence of the analysis was checked using Tracer v1.7.1 [41].

Next, two species-specific alignments were generated. For *B. taurus*, we included our eight newly sequenced *B. taurus* samples and all publicly available sequences for which we could determine the haplotype based on the same eight diagnostic SNPs used to assign our samples. The *B. primigenius*-specific alignment contained our samples as well as all publicly available sequences of European wild aurochs. A full list of public data used in our study is given in S3 Table. The two species-specific multiple sequence alignments were used to generate statistical parsimony networks using the R-script TempNet [42]. Networks we generated were not temporally separated, instead they separated our samples from publicly available data.

## Results

We were able to generate consensus sequences from 11 (out of the 18) bone samples that were based on a coverage between 3.34x and 72.93x. Those 11 were all found in caves. The remaining seven samples were discarded due to low mapping coverage and sequencing quality. The 11 consensus sequences covered between 44.88% (7,333 bp) and 99.99% (16,337 bp) of the cattle reference mitochondrial genome (GenBank accession number: AY526085.1). An overview over dating, mapping and SNP analysis is shown in Table 1. More detailed dating and mapping results, as well as a figure containing SNP positions are in S1 and S2 Tables, and in S1 Fig, respectively. All raw sequencing data is deposited in the NCBI SRA under BioProject ID PRJNA705960. Addtionally, the partial mitochondrial genomes are made public in NCBI GenBank under accession numbers MW689247-MW689255, except mitochondrial genomes of samples Bt_MR_(3736BP) and Bt_CO_(60BP), which cannot be made available at NCBI GenBank, since they do not meet the GenBank submission criteria (less than 50% ambiguous characters). The analysis of ancient DNA patterns using mapDamage [34] shows terminal transitions indicating cytosine deamination ranged from 0.25 to 0.67% in mean per sample. These numbers are reduced due to UDG-treatment of the libraries. Mean fragment lengths ranged from 48.05 to 53.23 bp, consistent with an ancient origin of the sequences.

Based on eight diagnostic SNP positions, the three *B. taurus* samples *Bt*_AR_(1200BP), *Bt*_AR_(2460BP), and *Bt*_SA_(1290BP)_2 could unambiguously be assigned to European haplogroup T3. The samples (*Bt*_SA_(1200BP)_1, *Bt*_AT_(3370BP), *Bt*_AT_(1600BP), and *Bt*_CO_(60BP)) displayed 3-6 SNPs characteristic for T3 but were missing data at the remaining diagnostic positions. The final sample *Bt*_MR_(3736BP) could not be assigned to a haplogroup with any confidence due to missing data at seven diagnostic positions. We did not detect evidence for any T1 (African descendant) haplogroups in our data. Haplogroup assignments based on the SNP analysis, results of dating, and further information about the samples are given in Table 1.

Tentative morphological identification of three of our samples as *B. primigenius* was confirmed by phylogenetic analysis: As shown in Fig 2, *Bp*_CL_(9100BP)_1, *Bp*_CL_(9100BP)_2, and *Bp*_CL_(9100BP)_3 grouped within the clade containing known *B. primigenius* mitochondrial genomes. The *B. primigenius* clade is a sister to a clade containing *B. taurus* haplogroups T and Q. This phylogeny also confirms all preliminary species assignments for our *B. taurus*

**Table 1. Summarized dating and mapping results for 11 samples of Galician cattle subfossils.** Samples are referred to by their sample code in the text, which includes information about the name of the cave in which the subfossil was found, the dating, and the species assignment. Context refers to the historical context based on dating; results of subfossil dating are based on carbon dating and calibration. Mapping depth and and the percentage of the reference genome covered give an overview of the mapping results. Species assignments are based on morphological and genetic data. The haplotype was defined based on SNP analysis.

| Sample | Cave | Context | Age cal BP (2-sigma) | Mapping depth | % of reference genome covered | Species | Haplotype |
|--------|------|---------|---------------------|---------------|-------------------------------|---------|-----------|
| Bt_SA_(1200BP)_1 | Saballeiros | High Middle Age | 1125 ± 119 | 18.39 | 93.08 | *Bos taurus* | T3 |
| Bt_SA_(1290BP)_2 | Saballeiros | High Middle Age | 1233 ± 59 | 72.93 | 99.99 | *Bos taurus* | (T3) |
| Bt_MR_(3736BP) | Mina Rubiais | Chalcolithic | 4082 ± 63 | 6.49 | 44.88 | *Bos taurus* | ? |
| Bt_AR_(1200BP) | Arcoia | High Middle Age | 1143 ± 88 | 30.95 | 98.75 | *Bos taurus* | T3 |
| Bt_AR_(2460BP) | Arcoia | Iron Age | 2535 ± 175 | 33.36 | 99.68 | *Bos taurus* | T3 |
| Bp_CL_(9100BP)_1 | Chan do Lindeiro | Mesolithic | 9295 ± 170 | 10.07 | 66.89 | *Bos primigenius* | P |
| Bp_CL_(9100BP)_2 | Chan do Lindeiro | Mesolithic | 9299 ± 167 | 41.34 | 99.51 | *Bos primigenius* | P |
| Bp_CL_(9100BP)_3 | Chan do Lindeiro | Mesolithic | 9216 ± 185 | 8.81 | 68.19 | *Bos primigenius* | P |
| Bt_CO_(60BP) | Chan do Lindeiro | Recent | 100% modern C | 7.24 | 45.99 | *Bos taurus* | (T3) |
| Bt_AT_(3370BP) | A Tara | Bronze Age | 3592 ± 101 | 17.78 | 94.83 | *Bos taurus* | (T3) |
| Bt_AT_(1600BP) | A Tara | Suevic kingdom | 1481 ± 87 | 8.98 | 63.25 | *Bos taurus* | (T3) |

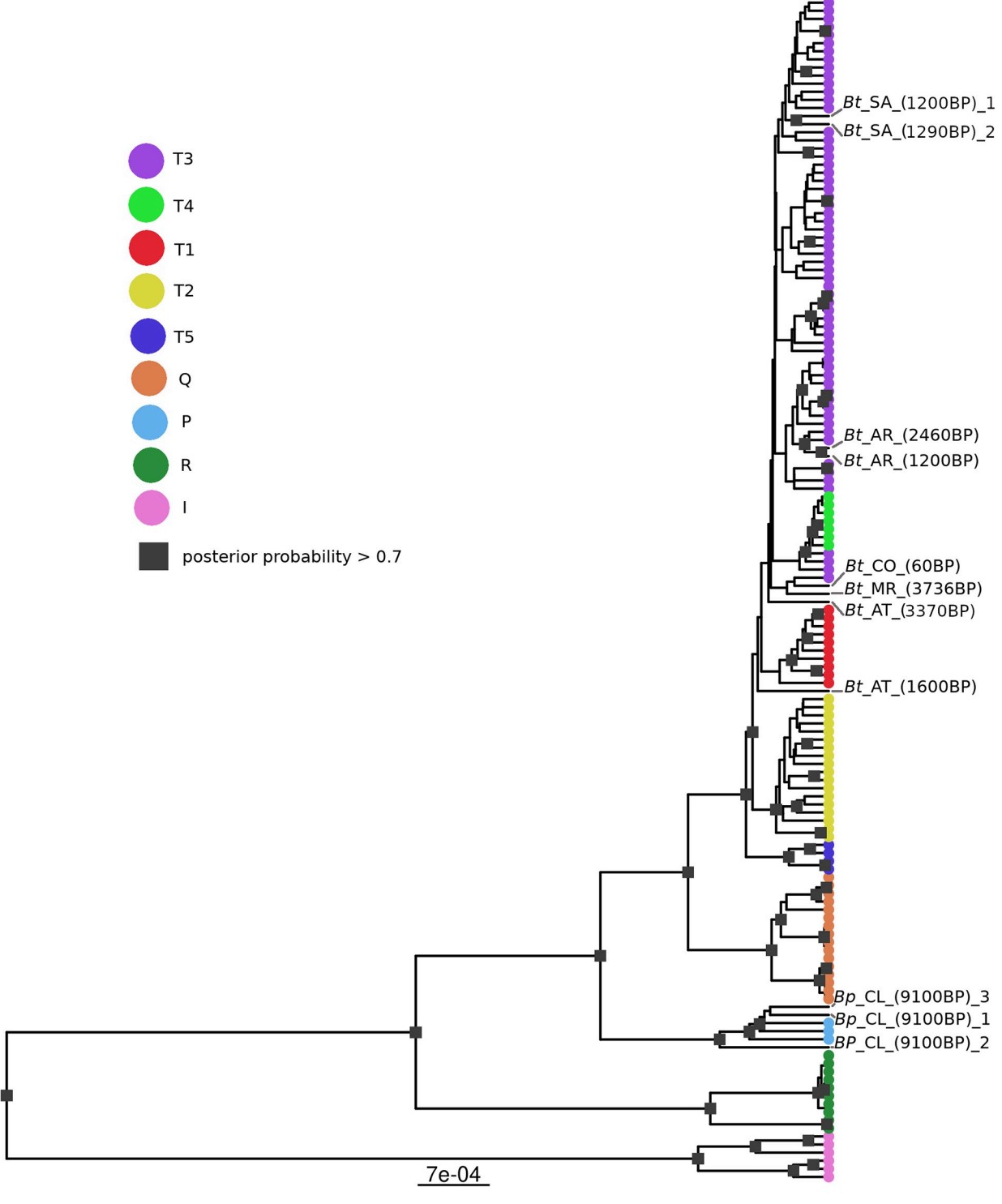

**Fig 2. Phylogenetic tree of mitochondrial genomes of different cattle species and haplogroups.** Result of a Bayesian analysis of mitochondrial genomes from five common cattle mitochondrial haplogroups (T,Q,P,R,I), which include the species *Bos taurus* (T,Q,R), *Bos indicus* (I), and *Bos primigenius* (P), and five subhaplogroups of T (T1,T2,T3,T4,T5). Reference mitochondrial genomes are coloured according to their haplogorup. Genomes generated in this study are denoted by their sample name. Squares on the nodes of the phylogeny mark a posterior probability larger than 0.7.

bone samples that were based on morphology. In the phylogeny, our *B. taurus* samples are assigned to a large and poorly resolved haplogroup T-specific clade that contains all sub-haplogroups that were included in this analysis, T1-T5. The samples from Arcoia are found within the T3 subclade of the large T clade, while all other samples are found either basal to the T3/4 clade or even basal to a clade of T3/4 and T1. Overall, this T clade shows only high support values at the branches that seperate T5 and T2 from the remaining T sub-haplogroups. The clade containing all *B. taurus* haplogroup R sequences is sister to a large clade containing the haplogroups T, Q, and P.

The statistical parsimony network that was computed from the *B. taurus*-specific alignment (Fig 3) shows a central hub representing mitochondrial genomes of different haplogroups. Most of these belong to T3, but all other haplogroups are present as well. Surrounding this central hub, nodes separate into haplogroup-specific satellite groups for T1, T3, and T. Where haplgroup T here refers to the sub-haplogroup T described in Troy et al., 2001 [3] and not to the general haplgroup. The samples *Bt*_AR_(2460BP), *Bt*_SA_(1290BP)_2, and *Bt*_AR_(1200BP) are deep within the T3 group. All other samples are closer to the central hub. Interestingly, our oldest *B. taurus* sample *Bt*_MR_(3736BP) was inferred as the central node of the central hub. However, the large amount of missing data in the sequence of this samples mitochondrial genome might influence its positioning in the network. The samples *Bt*_AT_(3370BP) and *Bt*_CO_(60BP) protrude into the sub-haplogroup T satellite group, and the remaining samples *Bt*_AT_(1600BP) and *Bt*_SA_(1200BP)_1 are in the outer ring of the central hub.

Compared to the *B. taurus* data, the statistical parsimony network of the *B. primigenius* samples shown in Fig 4 appears rather different. Sequences are separated in the network by overall more mutations than the *B. taurus* sequences. Our samples are older than published data and differ from other *B. primigenius* by at least ten mutations. This level of sequence divergence is comparable to that found in published data. Our sample *Bp*_CL_(9100BP)_1 appears as the the central node of the network, while all other nodes are surrounding this sample. The two other samples of this study are directly connected to the central sample, but separated from it by 15 and 22 mutations.

## Discussion

Ancient DNA analyses of domestic animals provide not only information about the history of these lineages and species, but also about the people who kept these animals. Along with the domestication of animals, there was a change in human societies to settlement and agriculture [2]. Applying phylogenetic analysis to mitochondrial genomes recovered from eleven cattle subfossil bones, we were able to confirm our tentative species assignments based on morphology. Samples identified as *B. taurus* group together with other, mainly modern, *B. taurus* samples, while our three *B. primigenius* samples form a clade with available mitochondrial genomes of other *B. primigenius*. Furthermore, the [14]C dating of our samples shows greater ages of the *B. primigenius* remains than for those of *B. taurus*, which is consistent with observations for other parts of Europe, where *B. primigenius* populations declined with the growth of human societies until their extinction in 1627 in Poland [43].

Results of our analyses allow insights into the time frame of Neolithization in Galicia. The samples that were identified as individuals of the wild aurochs *B. primigenius* were dated to around 9,000 BP (Table 1). Insight into the historic context of the age of these samples is given by comparison with the neighboring region Cantabria. The age of the *B. primigenius* samples dates to a period when the Mesolithic, the period before the Neolithic, just became common in Cantabria [25], and therefore to a time before the large transition to agricultural societies had

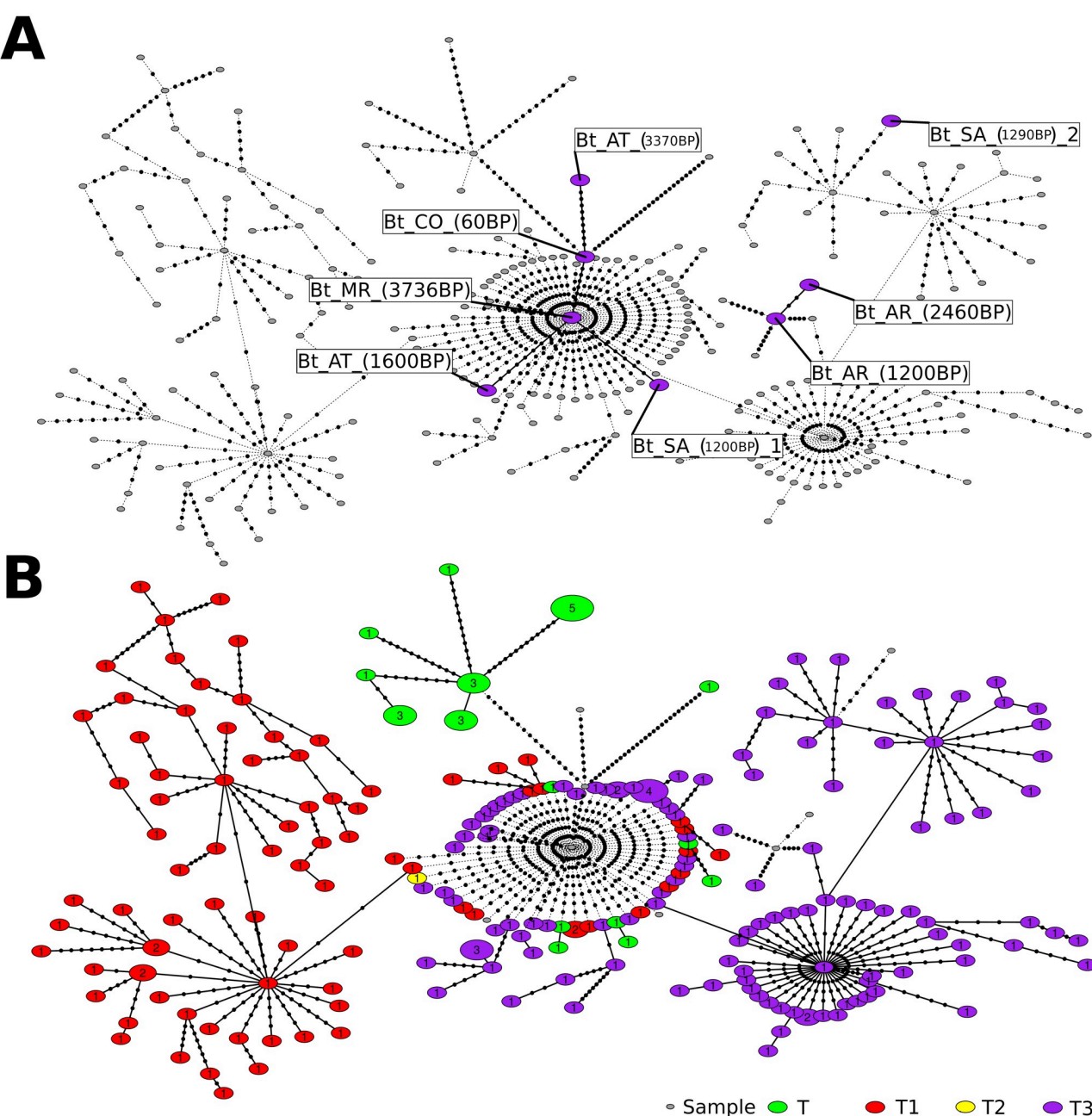

**Fig 3. Statistical parsimony network of *Bos taurus* mitochondrial genomes.** Statistical parsimony network of 243 *Bos taurus* haplogroup T mitochondrial genomes, including the four sub-haplogroups T, T1, T2, and T3. Circle sizes are proportional to the number of identical mitochondrial genomes in the node, and numbers inside nodes are exact numbers of mitochondrial genomes. Marks on the edges represent the number of mutations by which mitochondrial genomes differ from each other. Nodes that represent samples of this study are described according to their names shown in Table 1 and are colored in the upper panel (A) of the figure. The lower panel (B) shows colors for publicly available data. Colors correspond to haplogroups as determined by SNP analysis. Nodes corresponding to sequence data of the present study are colored as T3, although haplogroup assignment based on SNP is not certain for all of them (see text and Table 1 for details).

occurred. Unfortunately, our data set contains a large temporal gap, as the earliest *B. taurus* samples of this study date to 4082 ± 63, which, in the Cantabrian regions, is at the end of the Megalithic, the latest time period of the Neolithic [25]. Thus, based on our data we can conclude that by about 4000 BP, domestic cattle had arrived in the north of Spain, while 9,000

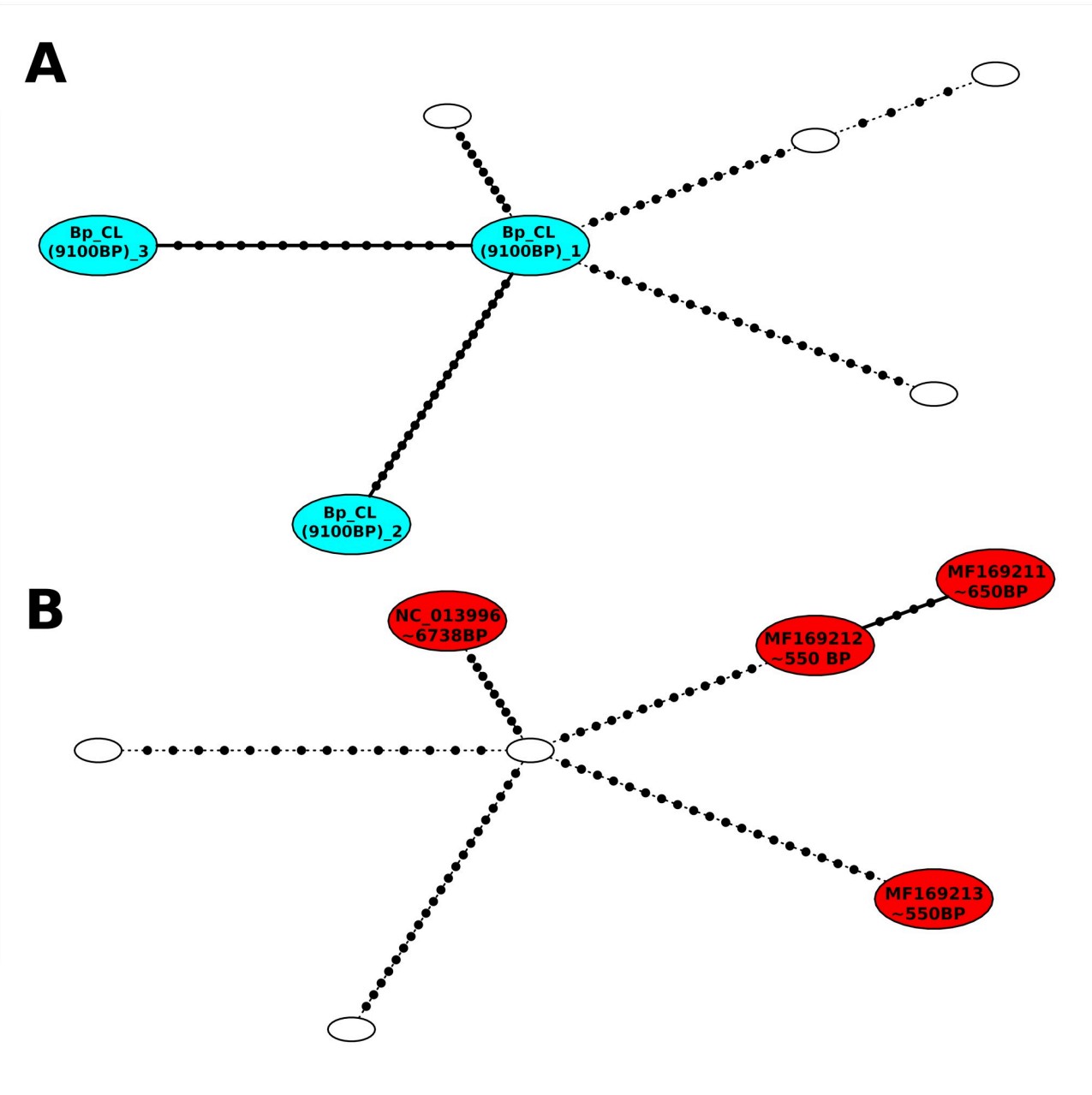

**Fig 4. Statistical parsimony network of *Bos primigenius* mitochondrial genomes.** Statistical parsimony network of seven *B. primigenius* mitochondrial genomes. Marks on the edges represent number of mutations by which mitochondrial genomes differ from each other. This figure includes three samples sequenced for this study, they are described by their names according to Table 1, and four published sequences, indicated by their NCBI accession number. Consistent with Fig 3, colors are used for the new samples of the current study in panel A and for publicly available data in panel B.

years ago, aurochs were still roaming the region. Determining when exactly in the intervening 5,000 years cattle were introduced and aurochs went extinct will have to await further archaeological findings.

Our results also provide insights into cattle populations from historical times. The study area was quite populated during the Iron Age, through a network of small fortified settlements

(castros) that were still active during the Roman conquest and settlement in the 1st century AD [44]. The Roman settlement in the Sierra was likely important for the empire as well as substantial, as evidenced by intensive exploitation of numerous gold mines in the area [45, 46]. Therefore, we would expect to find contributions from foreign livestock into the existing population. In many regions across Europe, the Roman occupation was followed by an increase in cattle size, a phenomenon that has been attributed to both the introduction of more robust animals and specific livestock management [47–49]. In Roman Galicia, an increase in the size of cattle has also been described [50]. If this increase in size was due to the introduction of foreign livestock, either the introduced animals were genetically indistinguishable from the local livestock, or only males were introduced, as we did not find any evidence of a genetic turnover of the mitochondrial lineages in the area. Further introductions of foreign livestock could have occurred during the period of migration (4th to 6th centuries), when successive Germanic peoples settled in ancient Roman Hispania: first the Suevi (409-585 AD), followed by the Visigoths, until the arrival of the Muslims at the beginning of the 8th century [51]. These peoples had been moving around the declining Roman Empire for more than a century. However, it is unlikely that they carried herds of cattle with them. Even less contribution is be expected from the Muslims conquerors of the Iberian Peninsula, as they never settled in Galicia [52]. Thus, it is not surprising that the cows of the Middle Ages studied here also do not reveal any genetic turnover compared to previous periods. However, those that are complete enough to be investigated with respect to their morphology (from A Tara cave) show a small size and dentition with strong mesowear, probably as a result of their life in the highlands [53]. Thus, it seems that it is the environmental conditions and the type of feeding that determined the size of historical Galician cattle, rather than any specific genetic lineage.

Maternal haplogroups of *B. taurus* provide information about cattle origin and migration, and they were historically defined based only a few diagnostic positions within the D-loop region of the mitochondrial genome [3, 6, 9, 36]. The network and Bayesian analysis of our study show that also whole mitochondrial genomes of this species group roughly according to those defined haplogroups (Figs 2 and 3). Our network analysis shows a very detailed picture of well resolved haplogroup clusters but also of an additional group of mitochondrial genomes in the center of the network that contains all different haplogroups and not only our ancient samples but also modern ones. Troy et al. describe the structure of haplogroups in this species as star-like, where the haplogroups T1-3 surround the haplogroup T, suggesting all other haplogroups coalesced from T [3]. Although the network analysis of this study shows a similar picture of haplogroups T1-3 surrounding a central group, that central group here contains a mixed of all haplogroups, with T forming a derived group from that. Our analysis, therefore, hints towards a more complicated origin and history of *B. taurus*.

For the samples sequenced in this study, the assigned haplogroups and the knowledge about their origin can be used to infer a possible descendance and migration of the cattle remains found in Galicia. In the SNP analysis, we detected only SNPs typical for sub-haplogroup T3 in our *Bos taurus* samples, which is common in Europe and Britain [3], although four of them have missing data in some of the diagnostic positions. A look at the analyses of their full mitochondrial genomes supports the hypothesis of their Europoean descent for the two samples from Arcoia and the sample *Bt_SA_(1290BP)_2*. In the phylogeny resulting from the Bayesian analysis they group clearly within in the T3 clade, which is also supported by the network analysis. In the our phylogeny, the sample *Bt_SA_(1200BP)_1* is placed in the T3 clade as well, but the network analysis placed it in the mixed group from which the haplogroups descend in a star-like formation. A European origin of this sample is, therefore, still likely but not as clear as for the samples mentioned previously. A more complicated picture becomes evident for the remaining samples. In the Bayesian analysis they are found to be basal

to the T3/4 clade, with one of them even basal to a T3/4 and T1 clade. However, the support of these branches is weak. In the network, the samples *Bt*_CO_(60BP) and *Bt*_AT_(3370BP) seem to be more closely related to genomes of the haplogroup T, which is the primarily domesticated haplogroup, and from which all others were derived, but which was was propagated into modern cattle [3]. This might be in accordance with the Bayesian analysis, in which this haplogroup is missing but must be assumed to be basal to the overall T clade. This is plausible for the older sample from A Tara, but not necessarily for the very young sample from Chan do Lindeiro. Its placement in the network, however, might be due to missing data in determining regions of the mitochondrial genome due to its low mapping coverage. Our sample *Bt*_MR_ (3736BP) is displayed in the very center of the network. Carbon dating identified it as the oldest of our samples, and it therefore might be more closely related to the founder population than to modern samples. However, the lack of all diagnostic SNPs and the overall lower coverage of this sample make an inference of the haplotype impossible and question its placement in the network. Regions in its mitochondrial genome which we could not characterize in this study can have a large impact on the results of all our analyses for this sample.

Our cattle haplogroup assignments are in agreement with ancient DNA data of human remains from Spain, which showed that the transition from Mesolithic to Neolithic societies was mediated by demographic processes, involving admixture with pioneering farmers of Near Eastern origin [23, 24, 54, 55]. This genetic signal of Neolithic immigration in humans is consistent with the first appearance of cattle carrying the T3 haplotype, likely introduced by the incoming farmers. Noticeable, although recent studies have shown prehistoric gene flow between human population from African and southern/central Iberia around the Middle Neolithic/Bronze Age [24, 56], we found no evidence of African diversity in the domestic cattle analyzed in this study, as none of them was related to the African haplogroup T1.

This study presents the oldest available sequenced mitochondrial genomes of *B. primigenius* to date. In the network analysis, one of our samples that is around 9100 years old is placed in the center of the network, from which not only the other two samples of this study connect, but also published samples (Fig 4). In light of the age difference between our samples and the oldest of the publicly available samples (6700 BP), this seems plausible. The statistical parsimony network of *B. primigenius* shows that individual mitochondrial genomes separate into distinct nodes that are separated by many mutational steps of samples of very different ages but also samples of similar age. Even the number of mutations separating the Galician samples is not very different from numbers of mutations between other samples. This suggests an overall higher genetic variability of *B. primigenius* mitochondrial genomes, which is likely because the domestication of *B. taurus* can be seen as a bottleneck through which this species went, caused by the initial domestication process, breeding, and selection by humans. Such a process did not happen in the naturally occurring population of wild aurochs, which spread across Europe and other continents. However, in comparison to *B. taurus*, there are very few sequenced mitochondrial genomes available for *B. primigenius*. Therefore, further sequencing and research of this lineage is necessary in order to resolve the historical population structure of *B. primigenius*.

## Conclusion

This study identified and characterized remains of 11 different Galician cattle roaming this region throughout different ages. Altogether, this adds to the picture of the history of different cattle species in Galicia, but, due to the close relationship of domesticated cattle with human, also to the history of human colonization and migration in that area. In accordance with human studies, we presented evidence that people with livestock migrated to that region from

the European mainland. Over centuries and many other human migrations, domesticated cattle in this region seem to have remained of European descent. The analyses of full mitochondrial genomes of cattle also gave insight in to the evolution of these animals in general. We showed that haplogroups defined based on only a small part of the mitochondrial genome were confirmed in analysis results of the full sequences. Additionally, these analyses confirmed the species assignment of the so far oldest sequenced mitochondrial genome of the extinct wild aurochs.

## Supporting information

**S1 Table. Dating information.** Table including detailed information about the dating of the samples as well as about the sites the samples were found.
(XLSX)

**S2 Table. Mapping information.** Table includes detailed information about the results of the sequencing and mapping experiments done with the samples of this study.
(XLSX)

**S3 Table. Publicly available data.** Table contains NCBI accession numbers of all publicly available data sets used for the analyses displayed in Figs 2–4.
(XLSX)

**S1 Fig. Diagnostic SNPs.** Image shows the the D-loop region of the mitochondrial genomes for all samples aligned. Only the bases in the diagnostic positions are displayed, other bases are displayed as "-".
(PNG)

## Author Contributions

**Conceptualization:** Stefanie Hartmann.

**Data curation:** Stefanie Hartmann.

**Formal analysis:** Stefanie Hartmann.

**Funding acquisition:** Aurora Grandal-d'Anglade.

**Investigation:** Marie Gurke, Amalia Vidal-Gorosquieta, Johanna L. A. Pajimans, Karolina Węcek, Axel Barlow, Gloria González-Fortes, Stefanie Hartmann, Aurora Grandal-d'Anglade, Michael Hofreiter.

**Methodology:** Marie Gurke, Amalia Vidal-Gorosquieta, Johanna L. A. Pajimans, Karolina Węcek, Axel Barlow, Aurora Grandal-d'Anglade, Michael Hofreiter.

**Project administration:** Stefanie Hartmann, Aurora Grandal-d'Anglade, Michael Hofreiter.

**Software:** Marie Gurke.

**Supervision:** Johanna L. A. Pajimans, Axel Barlow, Stefanie Hartmann, Aurora Grandal-d'Anglade, Michael Hofreiter.

**Writing – original draft:** Marie Gurke, Gloria González-Fortes, Stefanie Hartmann, Aurora Grandal-d'Anglade.

**Writing – review & editing:** Amalia Vidal-Gorosquieta, Johanna L. A. Pajimans, Karolina Węcek, Axel Barlow, Stefanie Hartmann, Aurora Grandal-d'Anglade, Michael Hofreiter.

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
