## [Decision Letter · Decision Letter 0]

29 Jan 2021

PONE-D-20-34180

Insight into the introduction of domestic cattle and the process of Neolithization to the Spanish region Galicia by genetic evidence

PLOS ONE

Dear Dr. Hofreiter,

Thank you for submitting your manuscript to PLOS ONE. After careful consideration, we feel that it has merit but does not fully meet PLOS ONE’s publication criteria as it currently stands. Therefore, we invite you to submit a revised version of the manuscript that addresses the points raised during the review process.

All comments need to be addressed before re-submission.

We look forward to receiving your revised manuscript.

Kind regards,

Peter F. Biehl, PhD

Academic Editor

PLOS ONE

Journal Requirements:

2.In your manuscript, please provide additional information regarding the specimens used in your study. Ensure that you have reported specimen numbers and complete repository information, including museum name and geographic location.

For more information on PLOS ONE's requirements for paleontology and archaeology research, see https://journals.plos.org/plosone/s/submission-guidelines#loc-paleontology-and-archaeology-research.

3. We note that you are reporting an analysis of a microarray, next-generation sequencing, or deep sequencing data set. PLOS requires that authors comply with field-specific standards for preparation, recording, and deposition of data in repositories appropriate to their field. Please upload these data to a stable, public repository (such as ArrayExpress, Gene Expression Omnibus (GEO), DNA Data Bank of Japan (DDBJ), NCBI GenBank, NCBI Sequence Read Archive, or EMBL Nucleotide Sequence Database (ENA)). In your revised cover letter, please provide the relevant accession numbers that may be used to access these data. For a full list of recommended repositories, see http://journals.plos.org/plosone/s/data-availability#loc-omics or http://journals.plos.org/plosone/s/data-availability#loc-sequencing.

4. Please amend the manuscript submission data (via Edit Submission) to include author Stefanie Hartmann.

5.We note that Figure(s) 1 in your submission contain map images which may be copyrighted. All PLOS content is published under the Creative Commons Attribution License (CC BY 4.0), which means that the manuscript, images, and Supporting Information files will be freely available online, and any third party is permitted to access, download, copy, distribute, and use these materials in any way, even commercially, with proper attribution. For these reasons, we cannot publish previously copyrighted maps or satellite images created using proprietary data, such as Google software (Google Maps, Street View, and Earth). For more information, see our copyright guidelines: http://journals.plos.org/plosone/s/licenses-and-copyright.

1) You may seek permission from the original copyright holder of Figure(s) 1 to publish the content specifically under the CC BY 4.0 license. 

Additional Editor Comments :

Please address the comments by reviewer 2 before re-submission.

Reviewers' comments:

Reviewer's Responses to Questions

**Comments to the Author**

1. Is the manuscript technically sound, and do the data support the conclusions?

Reviewer #1: Yes

Reviewer #2: Yes

2. Has the statistical analysis been performed appropriately and rigorously? 

Reviewer #1: Yes

Reviewer #2: Yes

3. Have the authors made all data underlying the findings in their manuscript fully available?

Reviewer #1: Yes

Reviewer #2: Yes

4. Is the manuscript presented in an intelligible fashion and written in standard English?

Reviewer #1: Yes

Reviewer #2: Yes

5. Review Comments to the Author

Reviewer #1: The paper "Insight into the introduction of domestic cattle and the process of Neolithization to the Spanish region Galicia by genetic evidence" provide new and useful data about cattle population dynamics in a non-sampled region of the Iberian peninsula. This data is important for the knowledge of the world cattle phylogeny and evolution as well as to understand ancient human history. The paper is very well written and developed and the analyses are appropriate and correct. I don't have any changes to highlight.

Reviewer #2: This work describes the retrieval of mtDNA genome sequence from 11 ancient Iberian Bos specimens. The data, as described, is of good quality; many of the genomes are of high coverage, the methods are sound, the laboratory track record is excellent. I think these data are of interest - three very old wild ox specimens are included for example, and the phylogenetic placing of some of the domestic sequences at basal phylogenetic positions is notable. Therefore my overall recommendation is for acceptance.

However I have a few comments, line numbers are given

20. the founding population estimate should be quoted as a range rather than the central estimate of 81 females.

table 1. the % coverage of three samples: Bt_MR, Bt_C, Bt_AT_1600BP and one Mesolithic are much lower than others. This needs to be borne in mind when one discusses the phylogenetic positions of these. For example is Bt_MR basal in position rather than simply incompletely described for several mutations that would remove it from this position?

174 relating to this point above, Bt_MR is inferred as the central node - but is it?

268, same point. Are we sure these are basal.

Fig 2, the BEAST tree. This analysis has the facility to consider nodes with different time horizons. A strong aspect of the data here is that these samples have direct C14 dates. I suggest that a better or additional analysis might only use the best covered mtDNA genomes and time-stamp the ancient samples. It would be interesting if the topology of the tree changed in this circumstance.

The discussion is somewhat lengthy and could be abridged. For example, I think that the relevance/comparison to the aDNA analysis of human remains is over-discussed. The only point (and it is worth making) is the absence of T1.

323 An archaeologist might query the claim that this presents “ the first evidence that people with livestock migrated into that region from the European mainland.” Is there no other non aDNA evidence for this?

Data availibility: It is welcome that all raw fastq data will be deposited. This is because processed data (eg mtDNA sequences) are subject to local parameter choices and subsequent users may wish to process these slightly differently.

6. PLOS authors have the option to publish the peer review history of their article (what does this mean?). If published, this will include your full peer review and any attached files.

Reviewer #1: No

Reviewer #2: **Yes: **Dan Bradley

---

## [Author Response · Author response to Decision Letter 0]

19 Mar 2021

Dear Editor,

thank you and the reviewers for the comments to our manuscript entitled “Insight into the introduction of domestic cattle and the process of Neolithization to the Spanish region Galicia by genetic evidence.” As requested, we have revised our manuscript and include a version with tracked changes. The point-by-point responses to the reviewers’ comments are below.

Editor Comments:

Response: We have used the most current version of the LaTeX template, available at https://journals.plos.org/plosone/s/latex. This template appears to be more recent than the listed PDFs. 

2. In your manuscript, please provide additional information regarding the specimens used in your study. Ensure that you have reported specimen numbers and complete repository information, including museum name and geographic location. If permits were required, please ensure that you have provided details for all permits that were obtained, including the full name of the issuing authority, and add the following statement: 'All necessary permits were obtained for the described study, which complied with all relevant regulations.' If no permits were required, please include the following statement: 'No permits were required for the described study, which complied with all relevant regulations.'

Response: All specimens studied for our manuscript are housed in the "Instituto Universitario de Xeoloxía, Universidade da Coruña", the director of which is one of the co-authors (Aurora Maria Grandal D'Anglade). Therefore, no permits were required, and we have added the corresponding statement to the manuscript. 

3. We note that you are reporting an analysis of a microarray, next-generation sequencing, or deep sequencing data set. PLOS requires that authors comply with field-specific standards for preparation, recording, and deposition of data in repositories appropriate to their field. Please upload these data to a stable, public repository (such as ArrayExpress, Gene Expression Omnibus (GEO), DNA Data Bank of Japan (DDBJ), NCBI GenBank, NCBI Sequence Read Archive, or EMBL Nucleotide Sequence Database (ENA)). In your revised cover letter, please provide the relevant accession numbers that may be used to access these data. For a full list of recommended repositories, see http://journals.plos.org/plosone/s/data-availability#loc-omics or http://journals.plos.org/plosone/s/data-availability#loc-sequencing.

Response: We deposited all raw sequence reads of this study in the NCBI Sequence Read Archive under BioProject accession number PRJNA705960. All generated mitochondrial genomes were deposited in NCBI GenBank. All accession numbers have now been included in the manuscript. 

The samples Bt_MR_(3736BP) and Bt_CO_(60BP) cannot be made available at NCBI SRA (really – you mean GenBank, or? SRA should be fine), since the do not meet the GenBank submission criteria (less than 50 \\% ambiguous characters). These are available on request. 

4. Please amend the manuscript submission data (via Edit Submission) to include author Stefanie Hartmann.

Response: The author Stefanie Hartmann is now included in the manuscript submission data. 

5.We note that Figure(s) 1 in your submission contain map images which may be copyrighted. All PLOS content is published under the Creative Commons Attribution License (CC BY 4.0), which means that the manuscript, images, and Supporting Information files will be freely available online, and any third party is permitted to access, download, copy, distribute, and use these materials in any way, even commercially, with proper attribution. For these reasons, we cannot publish previously copyrighted maps or satellite images created using proprietary data, such as Google software (Google Maps, Street View, and Earth). For more information, see our copyright guidelines: \\url{http://journals.plos.org/plosone/s/licenses-and-copyright}.

Response: The map was created by coauthor Aurora Grandal-d'Anglade specifically for this study; it has not been used elsewhere. We have added this information to the legend of Figure 1 in our manuscript.

6. Please include captions for your Supporting Information files at the end of your manuscript, and update any in-text citations to match accordingly. Please see our Supporting Information guidelines for more information: \\url{http://journals.plos.org/plosone/s/supporting-information}.

Response: We reorganized the data of our supporting information into several different files that follow the PLOS ONE supporting information guidelines. Captions and in-text citations in our manuscript were updated as requested. 

Reviewer 2 comments:

20. the founding population estimate should be quoted as a range rather than the central estimate of 81 females.

Response: We addressed this comment in line 20 of our manuscript. The sentence now reads: "Based on mitochondrial DNA data, a founding population of 29 to 783 females has been estimated for the beginning of the taurine cattle domestication in the Middle East [4, 17]"

table 1. the % coverage of three samples: Bt_MR, Bt_C, Bt_AT_1600BP and one Mesolithic are much lower than others. This needs to be borne in mind when one discusses the phylogenetic positions of these. For example is Bt_MR basal in position rather than simply incompletely described for several mutations that would remove it from this position?

Response: We agree with this comment, especially for the samples Bt_MR and Bt_CO. For both of them, we only were able to retrieve sequence information that covered less then 50% of the mitochondrial genome. Therefore, we added additional explanations for their placements in the phylogeny and the network in the discussion. Specifically, the following short sections were added.

line 271: "This is plausible for the older sample from A Tara, but not necessarily for the very young sample from Chan do Lindeiro. Its placement in the network, however, might be due to missing data in diagnostic regions of the mitochondrial genome due to its low mapping coverage."

line 277: "However, the lack of all diagnostic SNPs and the overall lower coverage of this sample make an inference of the haplotype impossible and question its placement in the network. Regions in its mitochondrial genome which we could not characterize in this study can have a large impact on the results of all our analyses for this sample."

174 relating to this point above, Bt_MR is inferred as the central node - but is it?

Response: This point is related to previous one, but is referring to a line in the results. Because in this line we only wanted to describe the samples placement in the network, we did not make any changes there. However, we do agree with the overall point that the central position of this sample might might be due to missing data at potentially important sections of the mitochondrial genome. 

268, same point. Are we sure these are basal.

Response: We addressed this comment specifically with the changes made in the discussion at line 271, shown two comments above. 

Fig 2, the BEAST tree. This analysis has the facility to consider nodes with different time horizons. A strong aspect of the data here is that these samples have direct C14 dates. I suggest that a better or additional analysis might only use the best covered mtDNA genomes and time-stamp the ancient samples. It would be interesting if the topology of the tree changed in this circumstance.

Response: We thank the reviewer for this suggestion and tried to implement it. Unfortunately, the BEAST analysis did no coalesce even after a very long run time (> 1 billion steps). Therefore, we assume that our data set does not contain enough temporal information for this type of analysis. Consequently, we have omitted it from the revisions.

The discussion is somewhat lengthy and could be abridged. For example, I think that the relevance/comparison to the aDNA analysis of human remains is over-discussed. The only point (and it is worth making) is the absence of T1.

Response: In order to abridge the discussion, we shortened the paragraph about the comparison with human remains. The new paragraph starts in line 282 of the manuscript and now reads as follows: 

"Our cattle haplogroup assignments are in agreement with ancient DNA data of human remains from Spain, which showed that the transition from Mesolithic to Neolithic societies was mediated by demographic processes, involving admixture with pioneering farmers of Near Eastern origin [23, 24, 44, 45]. This genetic signal of Neolithic immigration in humans is consistent with the first appearance of cattle carrying the T3 haplotype, likely introduced by the incoming farmers. Noticeable, although recent studies have shown prehistoric gene flow between human population from African and southern/central Iberia around the Middle Neolithic/Bronze Age [24, 46], we found no evidence of African diversity in the domestic cattle analyzed in this study, as none of them was related to the African haplogroup T1."

323 An archaeologist might query the claim that this presents “ the first evidence that people with livestock migrated into that region from the European mainland.” Is there no other non aDNA evidence for this?

Response: We agree with this comment, as well, and changed the concerned line 315 accordingly. It now reads: "In accordance with human studies, we presented evidence that people with livestock migrated to that region from the European mainland."

Data availibility: It is welcome that all raw fastq data will be deposited. This is because processed data (eg mtDNA sequences) are subject to local parameter choices and subsequent users may wish to process these slightly differently.

Response: The raw fastq data is deposited in the NCBI Sequence Read Archive under BioProject accession number PRJNA705960. Accession numbers have now been included in the manuscript. 

Other changes: 

While the manuscript was in review, we received new C14 dating information for three samples of this study. The concerned samples are Bt_AT_strat, Bt_SA_strat_1, and Bt_SA_strat_2. Their sample names were changed accordingly to Bt_AT_3370BP, Bt_SA_1200BP_1, and BT_SA_1290BP_2. Due to the new samples names, we made changes in Figure 2 and Figure 3 and also changed the names in the manuscript and Table 1. 

Yours sincerely, 

Marie Gurke, on behalf of all co-authors

---

## [Editor Report · Decision Letter 1]

22 Mar 2021

Insight into the introduction of domestic cattle and the process of Neolithization to the Spanish region Galicia by genetic evidence

PONE-D-20-34180R1

Dear Dr. Hofreiter,

We’re pleased to inform you that your manuscript has been judged scientifically suitable for publication and will be formally accepted for publication once it meets all outstanding technical requirements.

Kind regards,

Peter F. Biehl, PhD

Academic Editor

PLOS ONE
---

## [Editor Report · Acceptance letter]

5 Apr 2021

PONE-D-20-34180R1 

Insight into the introduction of domestic cattle and the process of Neolithization to the Spanish region Galicia by genetic evidence 

Dear Dr. Hofreiter:

I'm pleased to inform you that your manuscript has been deemed suitable for publication in PLOS ONE. Congratulations! Your manuscript is now with our production department. 

Kind regards, 

on behalf of

Dr. Peter F. Biehl 

Academic Editor

PLOS ONE